# Prevention Technology of Coal Spontaneous Combustion Induced by Gas Drainage in Deep Coal Seam Mining

Jiahui Li [1], Youxin Zhao [2] and Jinyu Du [3,*]

1 School of Civil Engineering, North China University of Science and Technology, Langfang 123000, China; lijiahui@ncist.edu.cn
2 Department of the Emergency Research Institute, China Coal Research Institute CCRI, Beijing 100013, China; zhaoyouxin@mail.ccri.ccteg.cn
3 Law School, Beijing Technology and Business University, Beijing 100048, China
* Correspondence: 20170901@btbu.edu.cn

**Abstract:** Due to high gas content and a low permeability coefficient in deep coal seam mining, the spontaneous combustion of coal around the wellbore can easily occur, leading to difficulties in extracting gas during the mining process. To determine the dangerous area around the borehole and conduct advanced prevention and control measures are the keys to preventing spontaneous combustion in boreholes. However, the dangerous area around the borehole is not clear, the sealing parameters lack scientific basis, and the key prevention and control measures are not clear, which have caused great harm to coal mines. This study took the 24,130 working face of Pingdingshan No. 10 Mine as an example, using numerical simulation, theoretical analysis, and field tests to classify the risks of studying the surrounding area of the wellbore. The dangerous area variations under different lengths of shotcrete in the roadway were analyzed, the optimal plugging parameters were studied, and the current "two plugs and one injection" plugging device was optimized. Based on the oxygen concentration and air leakage rate, a method was proposed to divide the dangerous area of fissure coal spontaneous combustion around the borehole induced by gas extraction. The dangerous area of spontaneous combustion around the borehole was defined as having an oxygen concentration larger than 7% and an air leakage rate less than 0.004 m/s. The comprehensive control measures of the grouting length at 2–4 m, hole-sealing parameter at 20-13 (hole-sealing depth 20 m, hole-sealing length 13 m) and the "two plugs, one injection and one row" device were determined.

**Keywords:** coal spontaneous combustion; hazard zone; roadway shotcrete; optimization of sealing parameters; two plugs; one injection and one row





## 1. Introduction

As of May 2021, there were 4536 coal mines in China; among them 62 mines with a depth of more than one kilometer have an average depth of about 1092 m, involving an approved production capacity of about 148 million tons. Studies have shown that coal mining in China is extending to deep coal mining at an average speed of 8–12 m/a. It is foreseeable that the proportion of deep coal mines will become larger and larger. The deep mining of coal resources will become the norm [1–6].

The occurrence environment of deep coal seam mining has the characteristics of "three high and one low": high ground stress, high ground temperature, high gas pressure, and low permeability. Therefore, mining efforts have been increased by increasing the drainage diameter, strengthening drilling, increasing negative pressure, and prolonging drainage time [7–9]. At the same time, the difference between deep coal seam mining and shallow coal seam mining is that the environment in which deep coal seams are located is a complex mechanical environment of "three highs and one disturbance". The comprehensive environmental factors around the gas drainage boreholes can easily cause spontaneous combustion of the surrounding broken coal bodies. Mine fires have always been a major

disaster that frequently occur in coal mine production and have serious consequences. During 2016–2020, roof and gas accidents are still the main major accidents with the largest number of gas accidents and fatalities in coal mines in the country, accounting for 18% and 62% of the total, respectively, as shown in Figure 1.

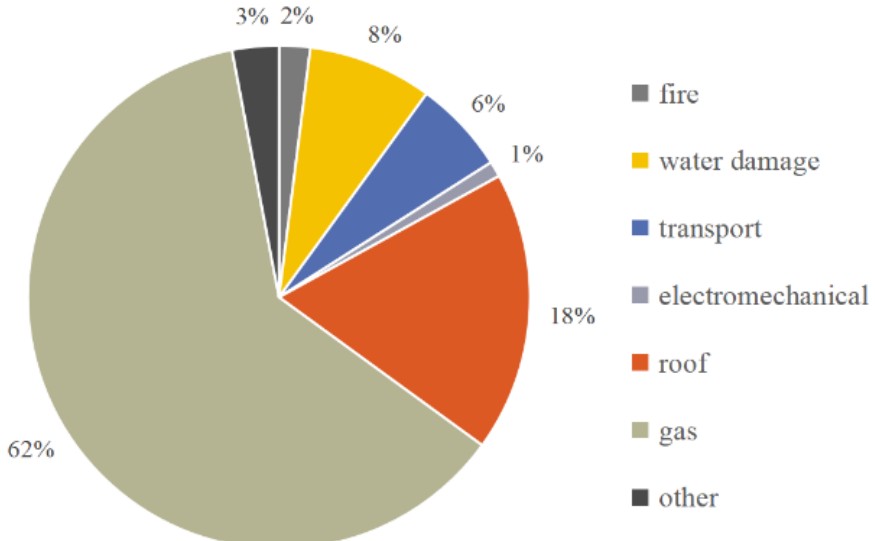

**Figure 1.** The number of deaths in various large accidents in coal mines.

The spontaneous combustion of coal bodies induced by gas drainage in deep coal seams is difficult to find and control. Determining the dangerous area will be the key to preventing spontaneous combustion in boreholes. Wang [10] proposed the use of infrared detection technology to measure the spontaneous ignition source range within a range of less than 10 m from the coal roadway. Wang [11] used a combination of theoretical analysis and engineering applications to establish a complete mathematical model, combined with a fuzzy comprehensive evaluation model of the loose coal particle size and the fuzzy judgment model of the spontaneous combustion hazard area of the roadway. Liang [12] used the gas concentration at some points to perform inversion using theoretical formulas and equations to obtain accurate dispersion coefficients and permeability coefficients, and use them as a basis to predict the location of the fire source. Qi [13] determined that the fire position of the gas drainage borehole was on the inner side of the blocked section based on wind speed, and conducted engineering experiments to verify each. Zhou [14,15] established a borehole spontaneous combustion model and proposed a quantitative evaluation index for gas drainage. Huang [16] studied the impact of periodical weighting by introducing weighting intervals into the coupled model of coal self-heating in the gob. Liu [17] mastered the air leakage law for the coal pillars on both sides and the gob, and determined the distribution law of the dangerous area of spontaneous combustion in the gob. Huang [18] completed a risk evaluation and located the spontaneous combustion in a fully mechanized gob.

The sealing depth and sealing length directly affect the sealing effect of the drilling. Liu [19] and others established the loose circle model of the surrounding rock of the rectangular roadway, studied the shape of the loose circle of the rectangular coal roadway and its formation and development mechanism, determined the loose circle theory formula for the reasonable sealing depth of gas drainage, and established a theoretical basis. Wang [20] studied the reasonable sealing depth of bedding boreholes using the cuttings method, and theoretical analysis showed that the reasonable sealing depth is between the boundary of the broken zone and the plastic zone of the surrounding rock. Xu [21] proposed a roadway excavation model about the intermediate principal stress and shear expansion of coal and rock and proposed a change of coal stress around the borehole and the expression of the borehole-sealing depth. Hao [22] used a fluid-solid coupling model to discuss the extraction

time, seal length, and the influence of air leakage on gas concentration. Wang [23] obtained a minimum seal length of 12 m from the ideal elastic-plastic model.

The plugging device is also an important part that affects the spontaneous combustion of coal around the borehole [24]. The plugging method commonly used in domestic coal mines is called "two plugging and one injection" [25–28]. The plugging device has the advantages of simple downhole plugging operation, fast plugging speed, and grouting enabled to penetrate deep into the fractures. However, because the "two plugs and one injection" device's two-capsule bags form a closed space during the grouting process, the internal gas pressure is too large to hinder the grouting of the grouting pump. When the internal and external pressures reach equilibrium, the gas in the grouting section cannot be completely discharged, and the slurry is not solidified, resulting in a reduction in the effective plugging length of the borehole.

In view of the various disasters caused by deep coal seams in the process of gas drainage, a single water injection method cannot fundamentally solve the problem of coal spontaneous combustion around the borehole. The study uses theoretical analysis, numerical simulation, and engineering applications to determine how to prevent the danger of coal spontaneous combustion around the borehole. Specifically, dividing the area, analyzing the influence of roadway shotcrete on the dangerous area around the gas drainage borehole, improving the "two plugs and one injection" plugging device, optimizing the plugging parameters, and scientifically predicting and adopting reasonable prevention and control measures to avoid disasters were determined. Safe and efficient gas drainage in deep coal seams is of great important for engineering practice.

## 2. Division of Dangerous Areas of Spontaneous Combustion in Coal Boreholes Induced by Gas Drainage

### 2.1. Analysis of Coal Spontaneous Combustion Area around Gas Drainage Borehole

A reasonable division of the dangerous areas around the gas drainage boreholes is the key to prevention, governance, and reform. In previous work [29–31], the authors conducted a detailed study on the cause, process, and influence of the natural combustion of the coal body around the borehole during gas drainage. However, at present, the dangerous area of coal spontaneous combustion around the gas drainage borehole is unclear and lacks theoretical basis.

As shown in Figure 2, the oxygen concentration of coal near the outer part of the plugging section is greater than the critical oxygen concentration of coal spontaneous combustion and the air flow speed is less than the critical wind speed of coal spontaneous combustion because the oxidized coal around the borehole is connected to the roadway, so an oxidation reaction occurs. The heat generated by the oxidation reaction is less than the heat dissipated at the place where the borehole and roadway are connected by gas extraction, and this area is a scattered zone. Beyond the scattered zone, along the direction of drilling, the oxygen volume fraction in this area decreases and the air flow velocity increases. The heat generated by coal oxidation is greater than the heat dissipated, and this area is the oxidation zone. Beyond the oxidation zone, along the direction of the drilling hole, oxygen concentration is reduced, airflow velocity increases, and heat dissipation is far greater than the heat release from oxidation. This area is the oxidation zone where the borehole fires spontaneously; inside the oxidation zone, the oxygen concentration is reduced, the wind velocity is high, heat accumulation in the area is difficult, and it is a suffocation zone. However, when the energy accumulated in the coal body around the borehole reaches a certain amount, the heat dissipation zone outside the plugging section has a low wind speed and high oxygen concentration, and the heat generated is far less than the heat dissipation. At this point, the heat dissipation zone would also be reduced and become a dangerous area for the spontaneous combustion of coal.

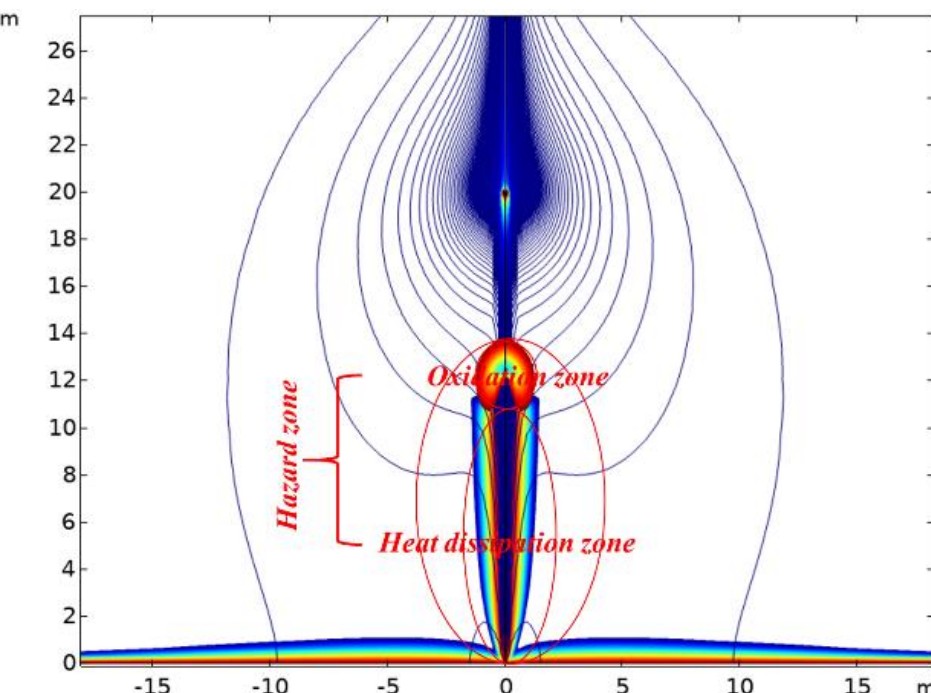

**Figure 2.** Diagram of the dangerous area around borehole.

### 2.2. Critical Oxygen Concentration and Critical Air Flow Velocity Determination

Coal spontaneous combustion occurs under three conditions: oxygen concentration, wind speed, and time [32–35]. Only when the ventilation and oxygen supply conditions are continuous and stable for a period of time, enough to combine the coal with sufficient oxygen, the oxidation process is fully carried out and the coal body can spontaneously ignite when a certain amount of heat is accumulated [36,37]. Therefore, choosing the appropriate critical oxygen concentration and airflow velocity is the key to classifying the spontaneous combustion of coal around the gas drainage hole. In order to obtain the critical oxygen concentration of coal spontaneous combustion, this study took the original coal sample from the 24,130 working face of Pingdingshan No. 1 Mine as an example, and conducted low-temperature oxidation experiments under the conditions of oxygen concentrations of 20.9%, 10%, and 8%. The relationship between the heat release and coal temperature and the specific experimental results are shown in Figure 3.

When oxygen concentration is 10.0%, the growth rate of CO and $C_2H_4$ output is inhibited, but there is still a large growth rate, indicating that when oxygen concentration reaches 10.0%, the spontaneous combustion of coal has been inhibited, but it is not enough to completely prevent the oxidation of coal. The inhibition of the coal oxidation process is stronger when the oxygen concentration continues to decrease to 8.0%. However, the first generation concentration of each gas behaves roughly the same. The change law of coal temperature reflects the degree of oxidation reaction during the spontaneous combustion of coal and the change curve is shown in Figure 3. It can be seen from Figure 3 that when the oxygen concentration is 20.9%, the oxidation process of coal has entered the intense oxidation stage; when the oxygen concentration is 10.0%, the coal sample also has an obvious violent oxidation process, but the violent combustion stage of the spontaneous combustion oxidation of the coal sample is obviously inhibited compared to the air condition; the coal temperature at 8.0% oxygen concentration is significantly lower than the coal temperature at 10.0% oxygen concentration, and the degree of oxidation is further suppressed. In summary, it can be obtained that the critical oxygen concentration of the experimental coal sample for spontaneous combustion is determined to be 8.0%.

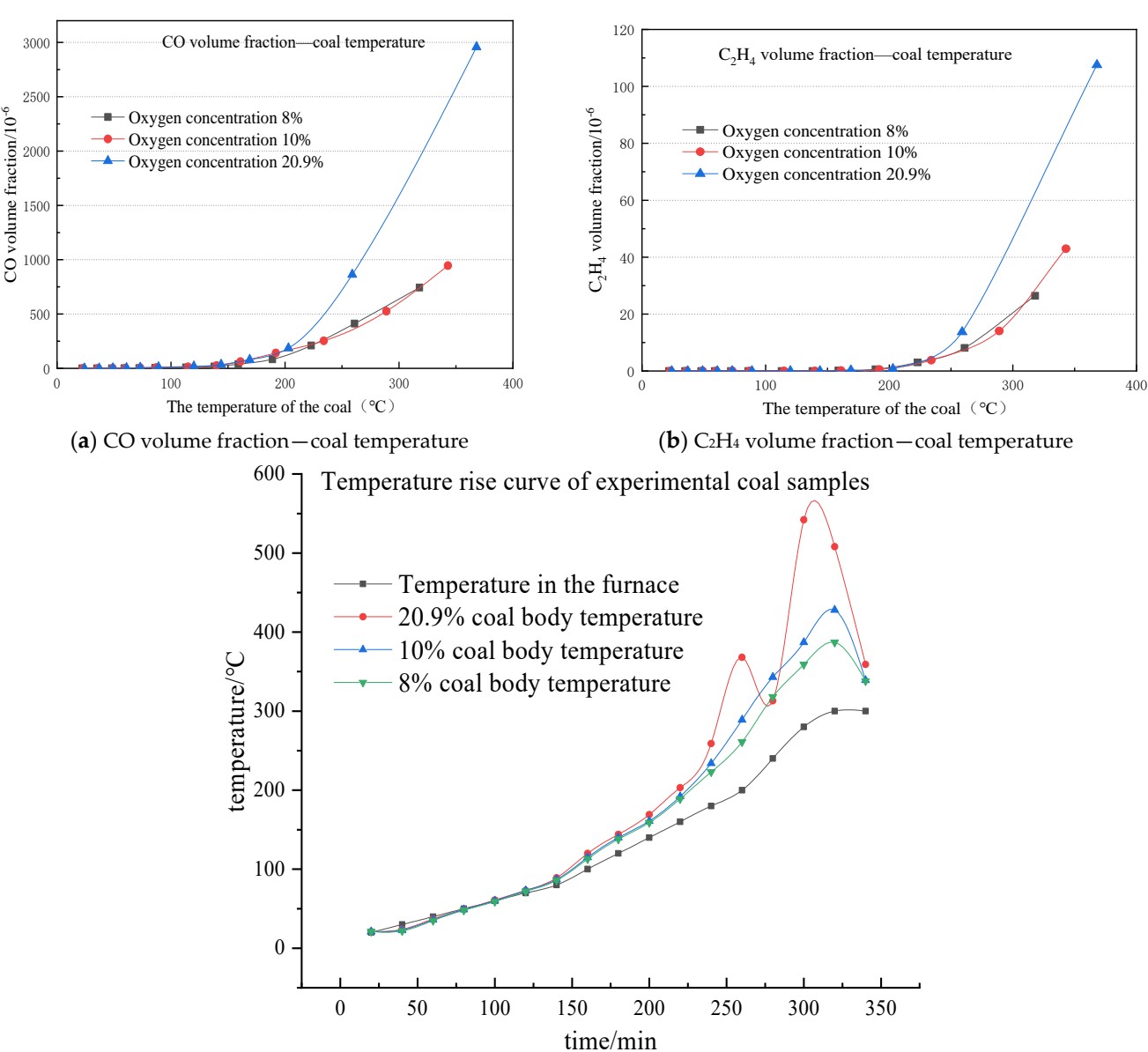

(**a**) CO volume fraction－coal temperature

(**b**) C₂H₄ volume fraction－coal temperature

(**c**) Temperature rise curve of experimental coal samples

**Figure 3.** The law of coal temperature, CO, and $C_2H_4$ concentration changing with time.

Air flow velocity is the main driving force for the spontaneous combustion and heat dissipation of coal around the borehole. When the airflow velocity of the gas around the borehole exceeds the critical velocity, even when oxygen is sufficient, it is difficult for heat to accumulate in this area. When the velocity is less than the critical velocity in this area, the amount of heat dissipation is less, and the spontaneous combustion of coal may occur after long-term oxidation heat release. Therefore, the critical air flow velocity is an important indicator to measure the spontaneous combustion of the coal around the borehole. Due to the complex environment of the coal body near the borehole, the wind flow velocity is difficult to measure, so this paper refers to multiple works (Wang H et al., 2002) to obtain a critical wind flow velocity of 0.004 m/s under the condition of sufficient oxygen.

### 2.3. Determination of the Hazardous Area

In the early stage of gas drainage, the heat dissipation zone is small and there is no risk of spontaneous ignition. In the later stage of gas drainage, the heat dissipation zone has the risk of coal spontaneous combustion. The area with spontaneous combustion is the

"dangerous area" and the suffocation zone is the "safe area". The safe area around the gas drainage borehole is characterized by a combination of an oxygen concentration less than 8% and wind velocity greater than 0.004, and the remaining areas are collectively referred to as "dangerous areas".

## 3. Key Technologies for Prevention and Control of Coal Spontaneous Combustion around Boreholes

In areas with serious borehole fires, measures are taken to stop drainage and inject water into the coal body area, but these measures can only temporarily control the spread of fire. Therefore, this thesis mainly discusses the prevention and control of spontaneous combustion in boreholes by using roadway shotcrete, optimizing hole-sealing parameters, and improving the "hole-sealing device".

### 3.1. Optimization of Roadway Shotcrete Length Parameters

Roadway shotcrete is the first line of defense that isolates the working face and the roadway, seals the cracks in the coal wall and the pores of the coal body itself, and reduces the interaction between the coal body and the air in the roadway. The length of roadway shotcrete affects the scope of the dangerous area around the borehole, so the method of combining numerical simulation and engineering is used to study the law of the influence of shotcrete on the oxygen concentration, fluid velocity, and the scope of the dangerous area around the borehole. According to the spontaneous combustion of the coal seam (Jia et al., 2021) in the 24,130 working face of Pingdingshan No.10 Mine, the numerical simulation set the plugging depth at 15 m, the plugging length at 8 m, the initial ground temperature of the coal seam at 48 °C, and the negative pressure of drainage at −23 kPa. For the model's parameters and boundary conditions, refer to Jia et al., 2021 and Zhao et al., 2021.

### 3.1.1. Analysis of Oxygen Concentration and Wind Velocity

Figure 4 is a cloud map of oxygen concentration and air flow velocity for different roadway shotcrete lengths during gas drainage for 3 days. Figure 4 shows that with the increase of roadway shotcrete length, the air flow velocity around the borehole shows almost no change at all and the area with a larger oxygen concentration changes a little. Oxygen is one of the factors that induces the spontaneous combustion of coal. Reducing the oxygen concentration around the borehole can effectively prevent gas drainage from inducing the spontaneous combustion of coal. The longer the distance of roadway shotcrete is, the greater the safe range of oxygen concentration around the borehole. It can be seen that when the roadway does not take roadway shotcrete measures, the characteristics of the wide range of oxygen concentration and the large oxygen concentration can easily cause the spontaneous combustion of coal in the fissures around the borehole during gas drainage. Because the roadway shotcrete has little effect on the air flow velocity, it can be obtained, from the perspective of oxygen concentration, that the roadway shotcrete can effectively prevent oxygen from entering the fractured coal around the borehole, thereby preventing the oxidation and exothermic reaction of the coal and reducing the possibility of coal spontaneous combustion.

### Analysis of Roadway Shotcrete in the Dangerous Area around the Borehole

Figure 5 shows that, without taking any measures, the longitudinal length of the dangerous area around the borehole is close to 14 m, in the shape of "hands clasped together", and the dangerous area is very large. When the shotcrete length of the roadway is 2 m, 4 m, and 6 m, the dangerous area of the fractured coal body around the borehole is reduced, and the range of the dangerous area near the roadway does not change much. When comparing different roadway shotcrete lengths, the hazardous area changes a little. Comprehensive analysis of the oxygen concentration, the fractured coal body around the borehole, and the hazardous area range, shows that the best roadway shotcrete length for a single borehole is 2 m.

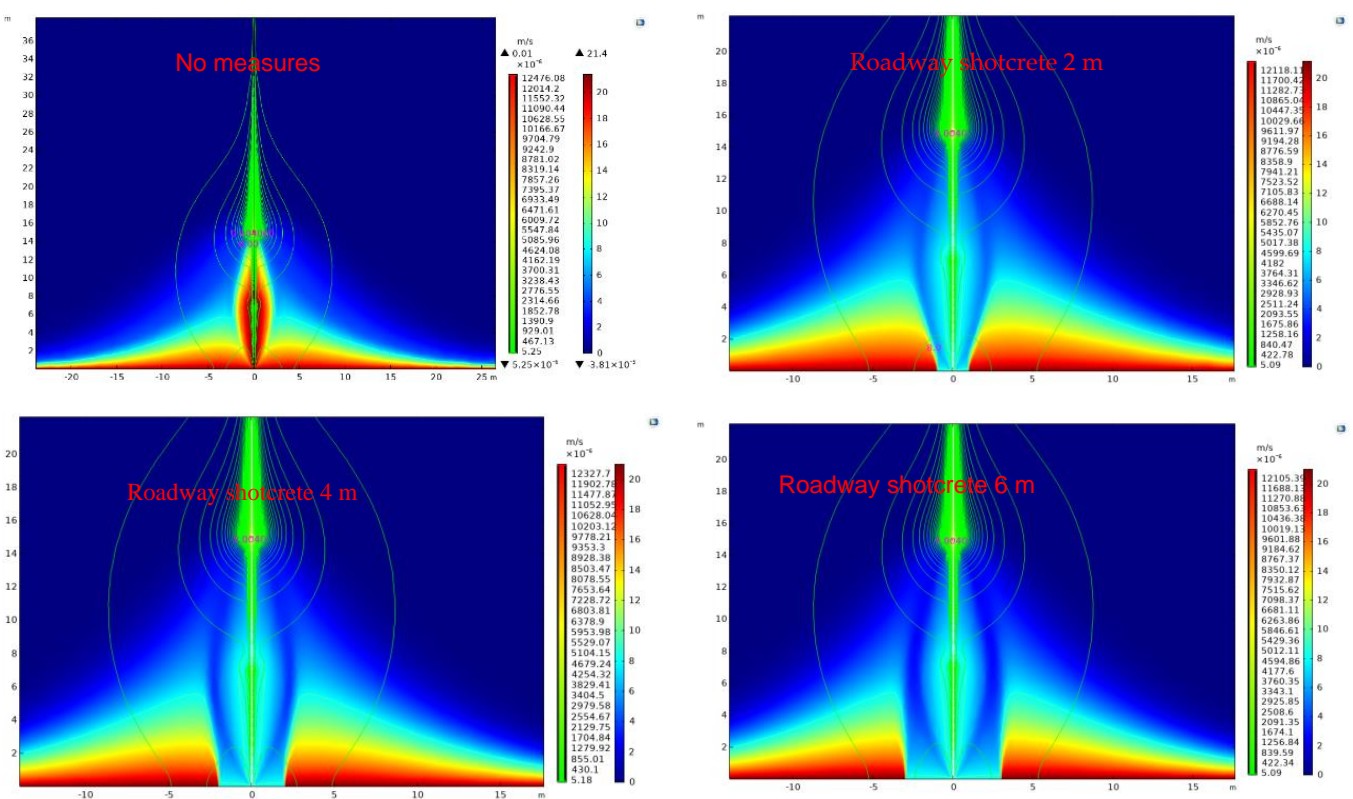

**Figure 4.** The cloud map of oxygen concentration for different roadway grouting lengths was obtained.

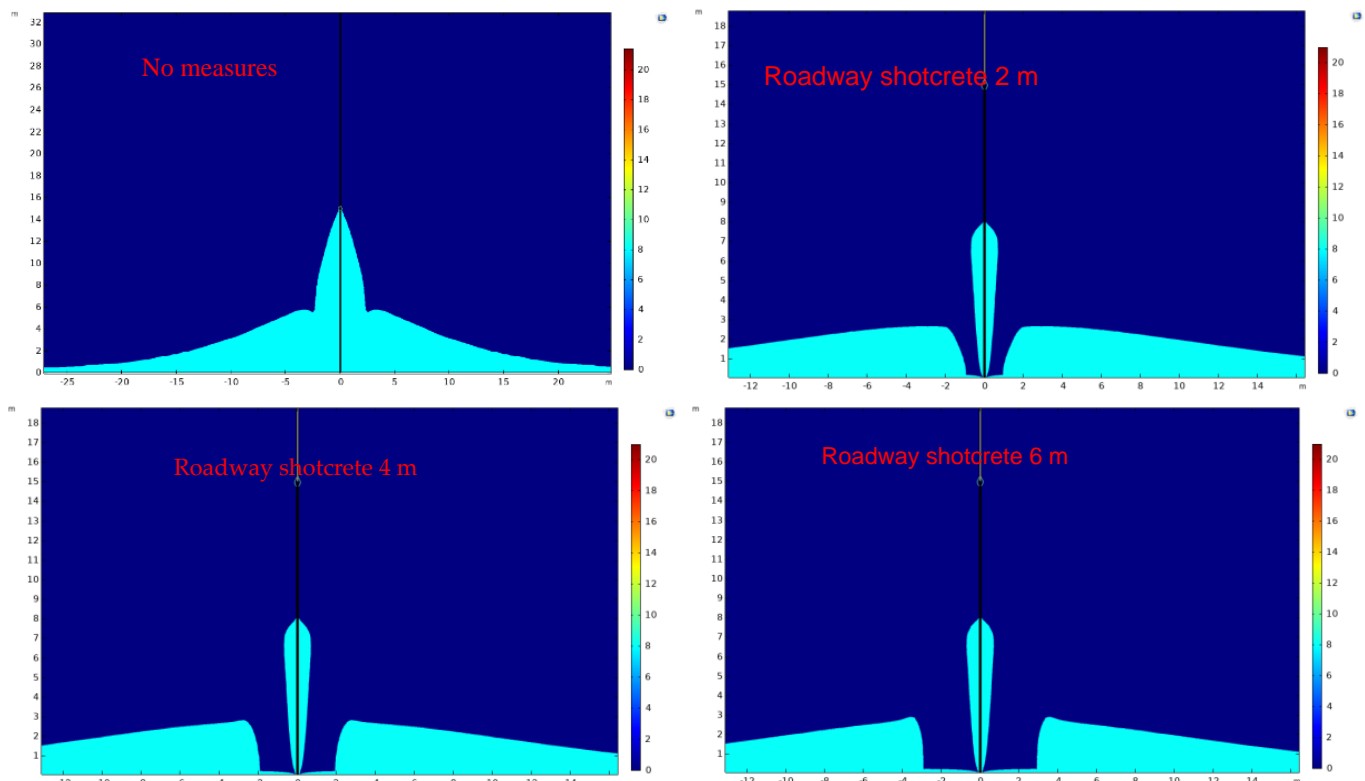

**Figure 5.** Changes of hazard areas under different roadway grouting lengths when the gas was extracted for 3 d.

3.1.2. Experimental Research on Shotcrete Engineering of Roadway

In the 24,130 working face, the tunnels around the experimental boreholes were cleaned by shotcrete, the spraying was even, and there was no "skirt wearing, barefoot" phenomenon; the spray thickness was 50 mm, and the experimental boreholes with shotcrete lengths of 0 m, 2 m, 4 m, and 6 m were set. Monitoring the CO concentration in the gas drainage borehole reflects the effect of roadway shotcrete. Figure 6 below shows the shotcrete on site and Figure 7 shows the monitoring result curve of the shotcrete length of different roadways.

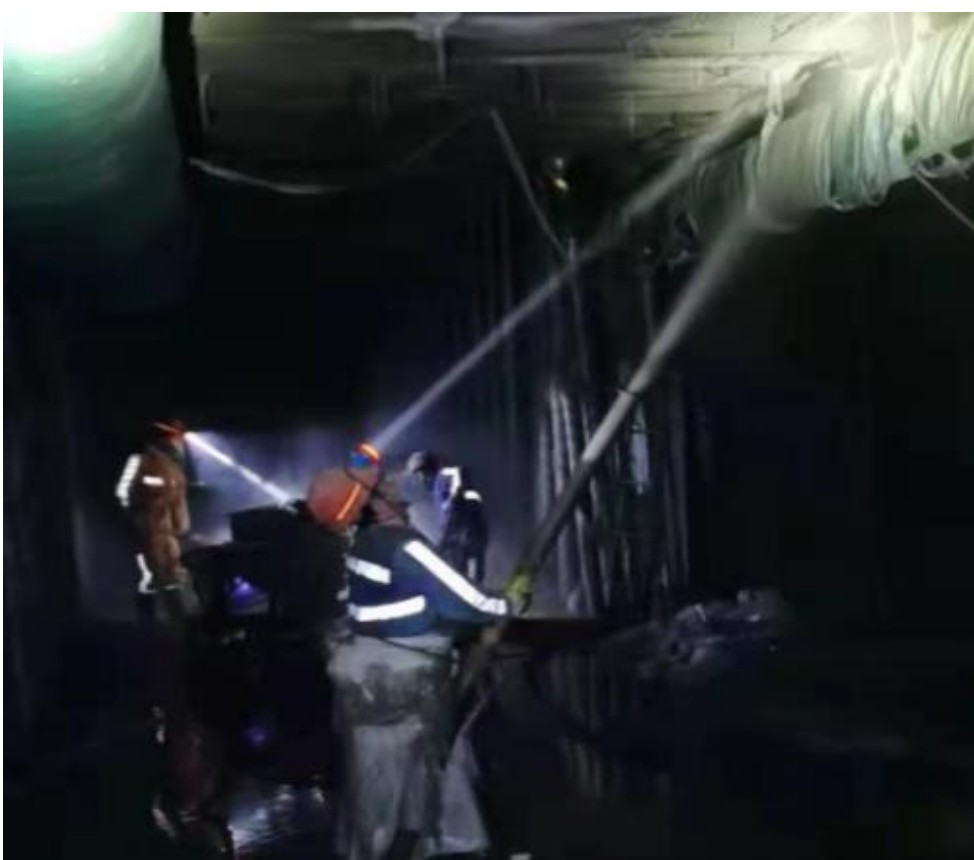

**Figure 6.** Site map.

The Figure 7 shows that when no measures are taken on the roadway, the CO concentration rises more obviously within 50 days, and the maximum can reach 430 ppm. When the roadway shotcrete length is 2, 4, and 6 m, the CO concentration around the borehole increases with the gas drainage. The increase in time continues, but compared to the 0 m roadway shotcrete, the rise tends to be slower. The average gas concentrations within 50 days of drilling gas drainage were 266, 210, 179, and 170 ppm for each shotcrete length, and the decline rate slowed down. Therefore, the length of the shotcrete for a single borehole is between 2 m and 4 m.

*3.2. Optimization of Drilling and Sealing Parameters for Gas Drainage*

One of the main reasons for spontaneous fire in gas drainage boreholes is the poor plugging effect of boreholes. The main factors that affect the plugging effect of boreholes are the length of the hole, the depth of the hole, the grouting material, and whether the plugging is standardized. To specific coal mines, reasonable plugging length and plugging depth are the most important, direct, and effective method to improve the plugging effect, which can not only greatly improve the efficiency of gas drainage, but also prevent spontaneous combustion in deep coal seam drainage boreholes.

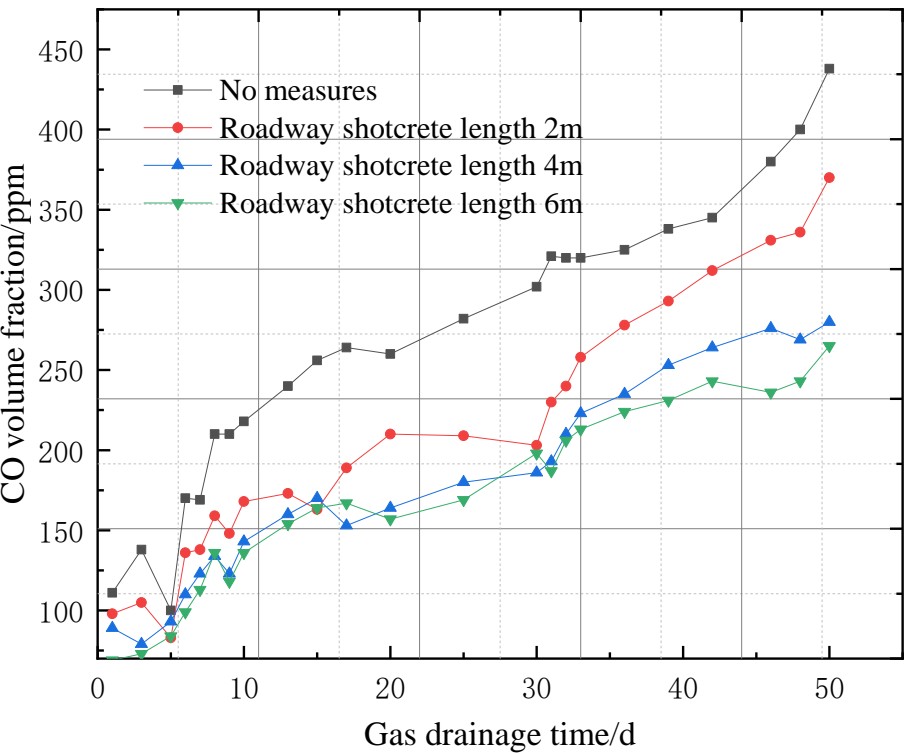

**Figure 7.** Results of testing.

3.2.1. Experimental Research on the Stress Distribution around the Roadway Based on the Drill Cuttings Method

The drill cuttings method is one of the effective methods used to test the stress distribution of the coal seam (surrounding rock). In field applications, pneumatic hand-held drilling rigs are generally used to drill holes perpendicular to the coal wall. The stress distribution characteristics of the coal body can be analyzed by the changing law of the amount of drill cuttings discharged from drilling and the accompanying dynamic phenomena.

The yield of drill cuttings changes with the change of stress in the coal body characterized by a certain functional relationship. As the stress of the coal body increases, the elastoplastic deformation increases and the yield of drill cuttings also increases. The drill cuttings method is used to study the characteristics of the stress distribution of the roadside coal body. The drill cuttings method studies the relationship between the pressure on the coal body and the yield of drill cuttings.

It can be seen from the formula that the yield of drill cuttings is positively correlated with the pressure of the surrounding rock. When the stress concentration increases, the quality of drill cuttings per unit length increases. Therefore, the stress state of the surrounding rock can be judged by the change in the yield of drill cuttings.

3.2.2. Determination of Sealing Depth

At the lower part of the 24,130 wind tunnel near the stop line, 594 mm drill holes were drilled into the coal tunnel along the layer, and drill cuttings were taken to test the stress distribution around the tunnel. Two 10 kg and 50 kg spring scales and two plastic buckets were used for on-site crumb fetching and weighing, as well as one square woven bag with a side length of about 1 m. The field test is shown in Figure 8.

The yield of drill cuttings from the 94 mm drill hole were plotted on a curve as shown in Figure 9. Figure 9 shows that the yield of cuttings in the first few meters of drilling does not change much, with only slightly increases. As the drilling depth increases, the yield of drill cuttings first rises sharply, then remains stable, then decreases, and finally stabilizes. Among them, 2#, 3#, 4# drill cuttings have a similar trend with the drilling distance. The 2# drilling hole enters the area where the yield of cuttings drops after the

yield of cuttings reaches the maximum. The 4# drilling hole has the largest yield of cuttings, indicating that the stress here is the largest. Combining the yield of drill cuttings in each drill hole, it can be seen that the yield of drill cuttings in the range of 0~7.5 m is low, with only small fluctuations. At this time, the yield of drill cuttings should only be the quality of the drilled coal body; drilling in the range of 8~17 m, the yield of cuttings increases sharply and remains high in the range of 17~23 m. It decreases sharply in the range of 23~28 m and tends to be stable after about 28 m. At the same time, in the range of 8~28 m, there were different numbers of coal cannons during the drilling process and the sound was loud or quiet, accompanied by the phenomenon of spray holes, the particle size of coal cuttings becoming larger, and the maximum yield of drill cuttings appearing in the drilling hole at a distance of 20 m before and after. Combined with the on-site measurement of the stress distribution of the surrounding rock using the cuttings measurement method and the theoretical analysis of the loose zone, it can be concluded that 0~7 m is the broken zone, 8~19 m is the plastic zone, and 20~28 m is the elastic zone. The maximum yield of drill cuttings appear around 20 m and the maximum stress appears at about 20 m away from the roadway.

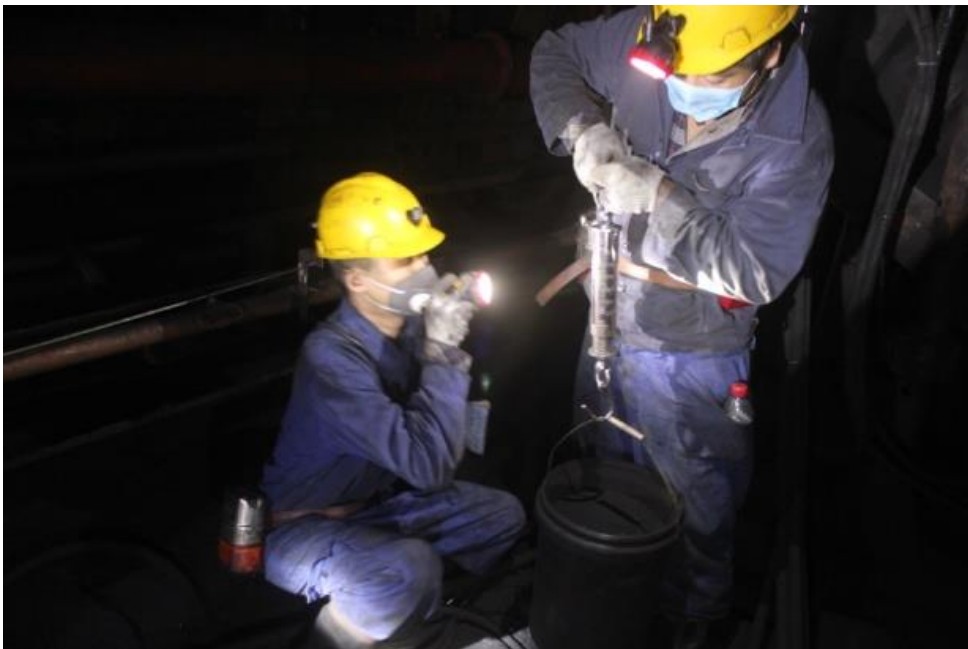

**Figure 8.** Field pictures.

### 3.2.3. Determination of Sealing Length

According to the actual situation of the sealing parameters of Pingdingshan No.10 Mine, the sealing length is set to 8 m, 10 m, 13 m, and 15 m. For other physical parameters and boundary conditions, please refer to the paper by Jia et al., 2021; Zhao et al., 2021. The coal temperature cloud map at 90 days around the borehole is shown in Figure 10.

Figure 10 shows the plugging depth of 20 m; the plugging lengths are 8 m, 10 m, 13 m, and 15 m, the gas drainage is 90 d, and there is coal temperature distribution around the borehole. It can be seen that when the sealing length is 8 m, 10 m, 13 m, and 15 m, the high temperature of the coal body around the borehole is 83.4 °C, 65.5 °C, 53.9 °C, and 48 °C, respectively, and the oxygen concentration around the borehole is greater than 8%. The area gradually decreases as the sealing length increases. From the perspective of temperature analysis, when the gas drainage is within 90 d, the sealing hole length should be no less than 10 m; when the gas drainage is greater than 90 d, the sealing hole length should be greater than 13 m. However, according to the gas drainage rules, it can be obtained that the working face gas drainage should last longer than 6 months, so the best sealing length is 13 m. From the perspective of oxygen concentration, although the area where the oxygen

concentration is greater than 8% is the smallest when the sealing length is 15 m, considering that the sealing length is too long, it also increases the difficulty and cost of sealing and when the sealing length is 13 m, the area where the oxygen concentration is greater than 8% is still small. Combining temperature changes and oxygen concentration changes, the optimal sealing hole length is 13 m, which can prevent the spontaneous combustion of coal seams and ensure efficient gas drainage.

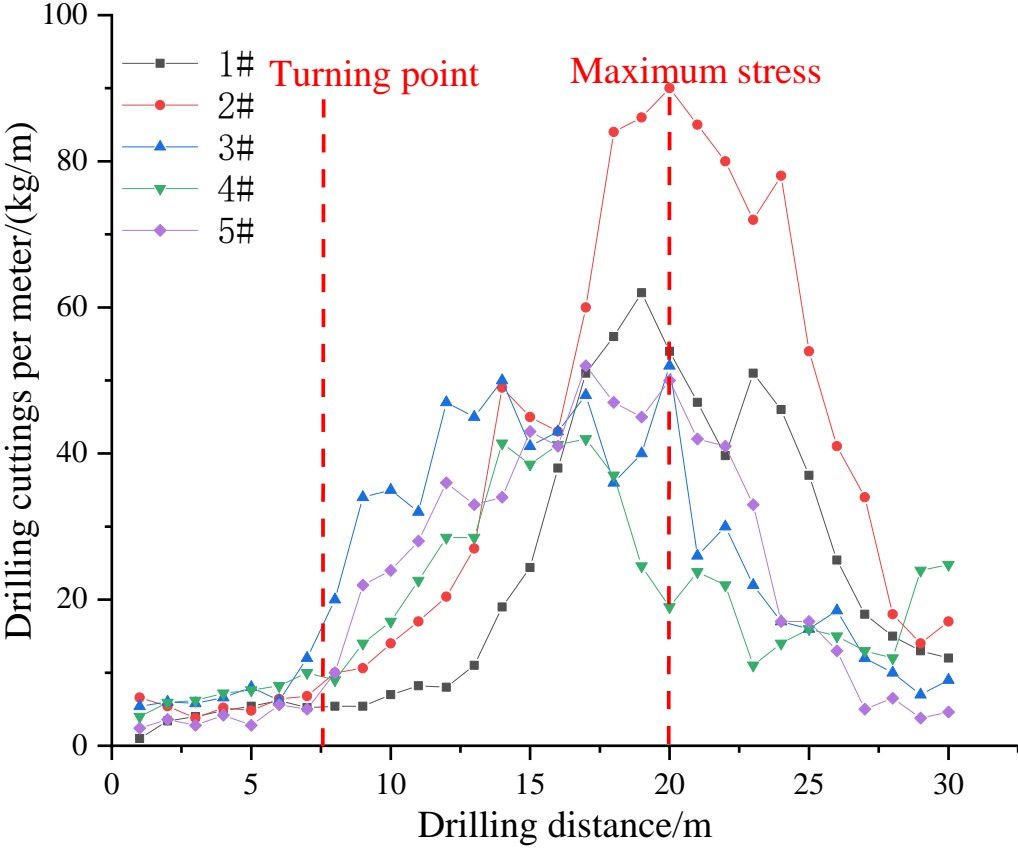

**Figure 9.** Variation of drilling yield with drilling depth.

The test was carried out on the lower edge of the 24,130 wind tunnel at a distance of 500 m from the cut hole. The test involved parameters of (sealing depth-sealing length) 15-13, 18-13, 20-8, 20-10, and 20-13. Concerning drilling hole parameters, each group of drilling spacing is 2 m, the hole diameter is 94 mm, and the drilling depth is 90 m. There were five holes in each group of boreholes. Each borehole in each group was connected to the branch pipe and then connected to the main pipe. Each group of boreholes had a monitoring hole to monitor the average gas of each group of boreholes. To extract the volume fraction, the average gas extraction volume fraction of each group of extraction boreholes was monitored once a day and recorded and the six groups of extraction boreholes were continuously monitored for 60 days. The data are plotted in Figure 11.

Comparing the sealing parameters of 20-8, 20-10 and 20-13, the average gas extraction volume fraction of boreholes with a sealing parameter of 20-8 is lower and the average gas concentration is 41%; when the sealing parameter is 20-10, the average gas drainage concentration accumulated within 50 days is 56.4% and when the drainage parameters are 20-13, the average gas drainage concentration reaches 70.8%. All comparative analysis shows that when the sealing depth is 20 m, the best gas drainage length of the sealing hole is 13 m. From the perspective of gas drainage concentration, it can prevent spontaneous combustion of the borehole. Comparing the three groups of holes with sealing parameters 15-13, 18-13, and 20-13, the average gas concentration of the drainage holes is 56%, 62.4%, and 70.8%, respectively, and the sealing length is the same, the deeper the sealing depth,

the higher the gas drainage concentration. When the 50 d plugging depth is 20 m, the average gas drainage concentration reaches 70.8%, indicating that the plugging effect is better and there is almost no air leakage. The experimental analysis of the drill cuttings method results in the peak stress of 24,130 working face. The point is 20 m and the sealing depth is increased before the stress peak, which can improve the extraction effect.

Based on the above analysis, the plugging depth is 20 m and the inner end of the plugging section of the plugging position is just at the peak stress point, which will not cause a blind area for drainage. When the plugging length is 13 m, which can prevent the natural fire of the coal seam and ensure the efficient gas drainage pick. Therefore, in order to increase the extraction efficiency and avoid the spontaneous combustion of the fractured coal around the borehole, comprehensive numerical simulation and field engineering experiment analysis can determine the optimal sealing parameters of the coal seam at the 24,130 working face as 20-13.

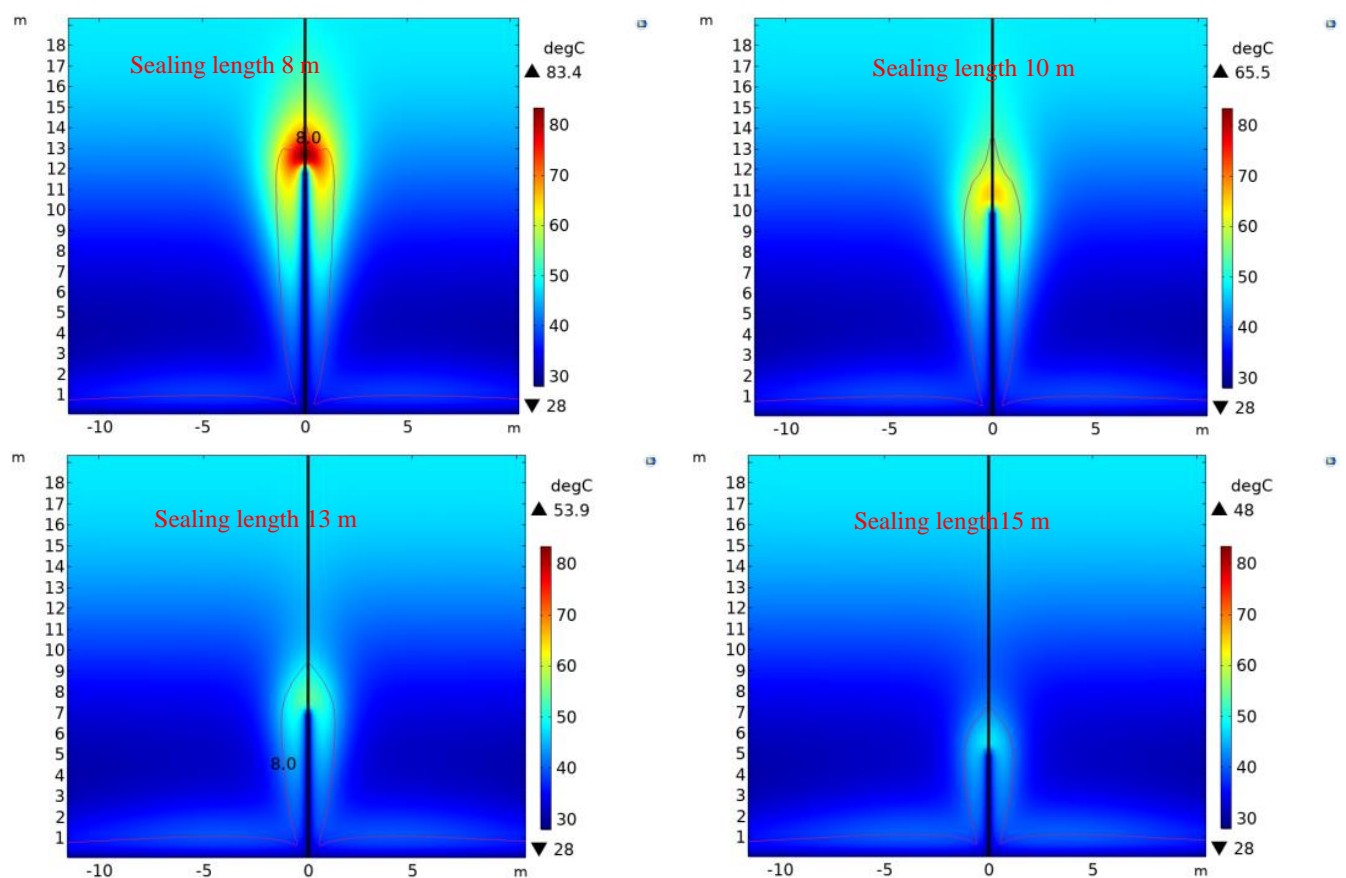

**Figure 10.** The temperature cloud map of the coal body around the borehole.

### 3.3. New Prevention Technology of "Two Plugs, One Injection and One Row"

The most commonly used downhole "two plugs and one injection" hole plugging device is used to observe the pressure with a grouting pressure gauge, which does not indicate that the borehole is filled with slurry. The gas in the borehole cannot be completely discharged, and the slurry can easily gather around the borehole after solidification, causing the formation of gaps and gas segments.

### 3.3.1. Development of "Two Plugs, One Injection and One Row" Device

The "two plugs, one injection and one row" plugging device is composed of three main parts: inside and outside sealing pockets (two plugs), grouting pipe (one injection), and grouting pipe (one row); it is made with PVC-U special glue. After connecting the suction tube with glue, the outer sealing bag and the inner sealing bag were tied in a fixed

position so that the suction pipe passes through the outer and inner sealing bags and the grouting pipe passes through the outer sealing bag. The bag and the inner sealing bag are parallel to the axis of the extraction pipe; the grouting pipe is located in the pipeline inside the outer sealing bag and the inner sealing bag. A one-way valve is provided in the pipeline and the pipe wall. There is a grout outlet on the grouting pipe so that the grout flows into the outer and inner sealing bags after the grout passes through the one-way valve in the grouting pipe; when the upward drilling hole sealer is arranged, the adjustment row, the distance from the position of the slurry pipe to the inner sealing bag, is 5~10 cm and the position of the blasting valve to the mouth of the slurry pipe is 20~30 cm, as shown in Figure 12a; when the downward drilling and sealing device is arranged, the distance between the position of the discharge pipe and the outer sealing bag to 5~10 cm were adjusted. The position of the blasting valve to the mouth of the discharge pipe is 20~30 cm, as shown in Figure 12b. The grouting process operates as shown in Figure 13.

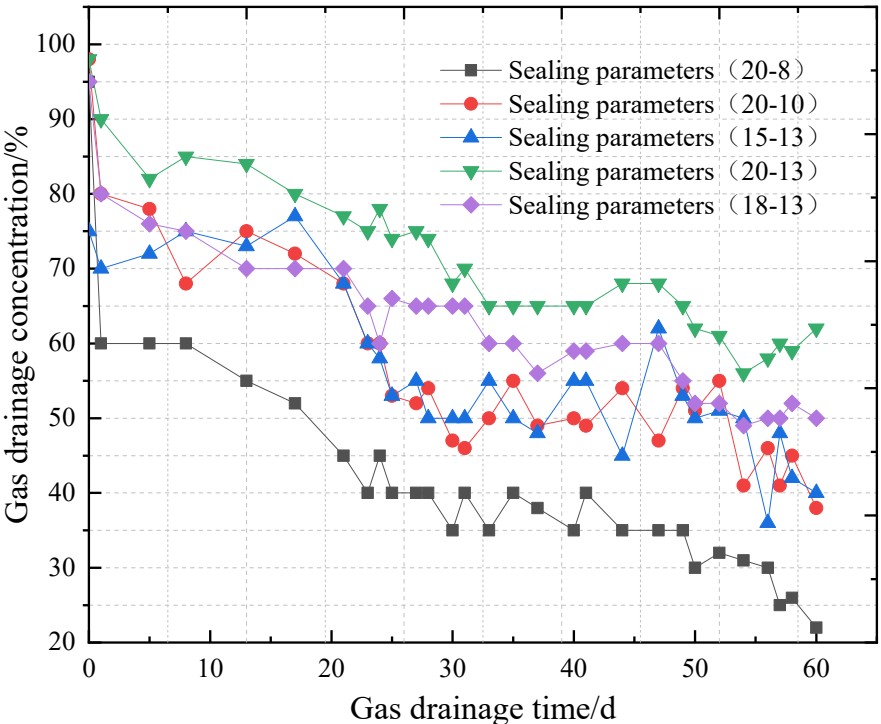

**Figure 11.** Average volume fraction of gas extraction from boreholes.

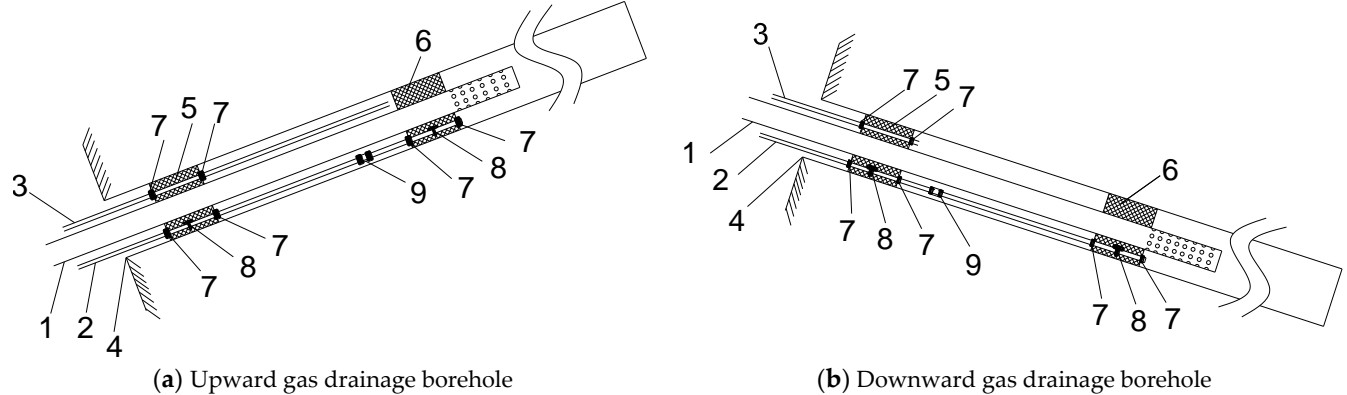

(**a**) Upward gas drainage borehole                                    (**b**) Downward gas drainage borehole

**Figure 12.** Schematic diagram of structures of hole packers for upward boreholes. 1 Extraction pipe; 2 grouting pipe; 3 grouting pipe; 4 mine drilling; 5 external blocking bag; 6 inner blocking bag; 7 metal clamp; 8 one-way valve; 9 burst valve.

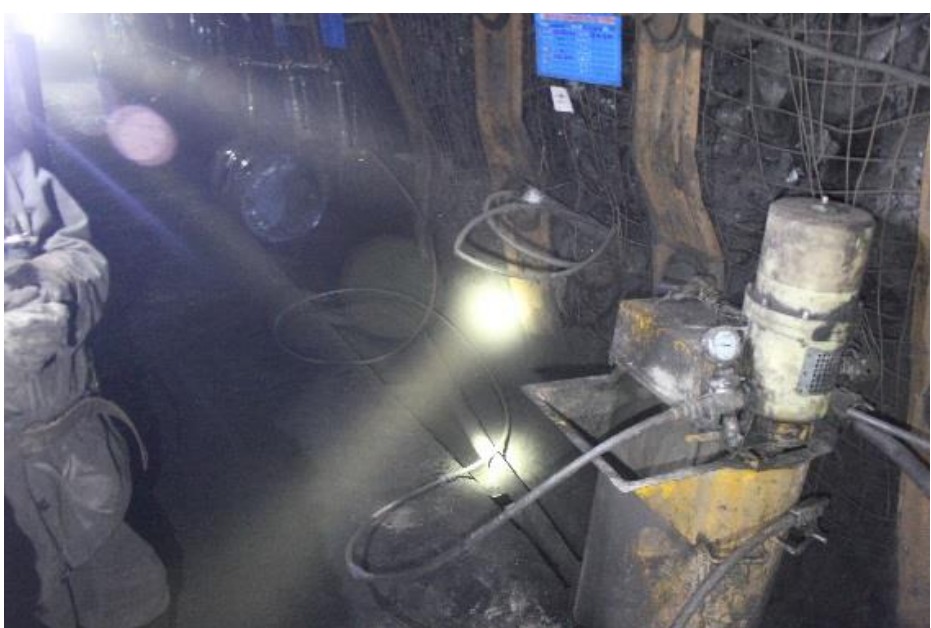

**Figure 13.** The entity of grouting.

3.3.2. Analysis of the Effect of the "Two Plugs, One Injection and One Row"
Plugging Device

(1) Drilling and cutting analysis

The simulated drilling adopts a seamless steel pipe (specification is 12 m/piece,
100 mm diameter), the extraction pipe adopts a polyvinyl chloride pipe (specification is
3 m/piece, plug-in type, 50 mm bore diameter), and the hole sealer is designed to be
8 m long and 2 pcs. The length of the capsule band is 800 mm. Two sets of airtightness
comparison tests were designed, using "two plugs and one injection" for up- and down-
holes and "two plugs, one injection and one row" for up- and downholes. The cutting
machine separately cut two simulated drill holes near the capsule belt near the exhaust pipe.
Whether the grouting is full is judged by whether the section is compact and complete, and
the cross-sectional views are shown in Figures 14 and 15.

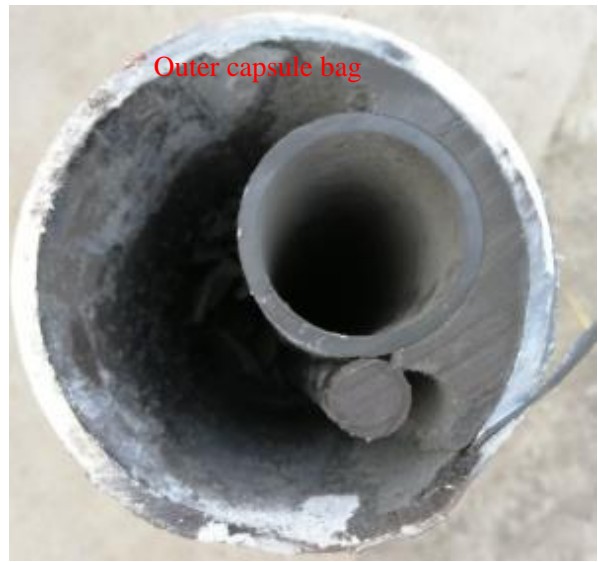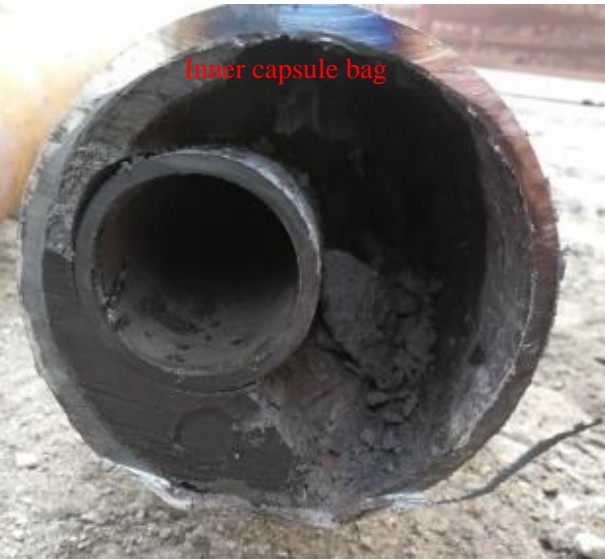

**Figure 14.** Borehole profile of boreholes for the "Two plugs and one injection" device.

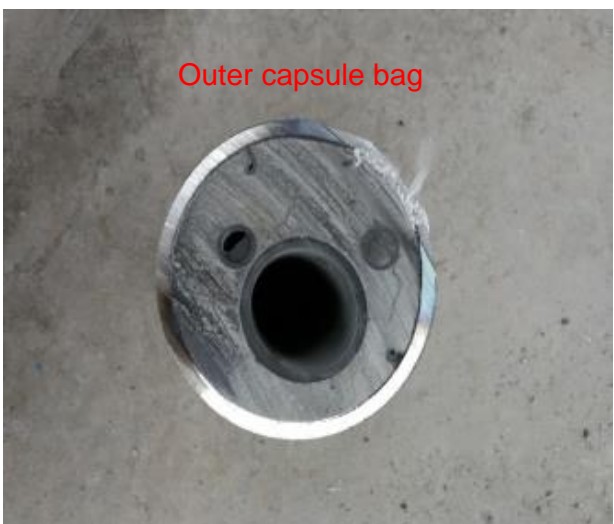
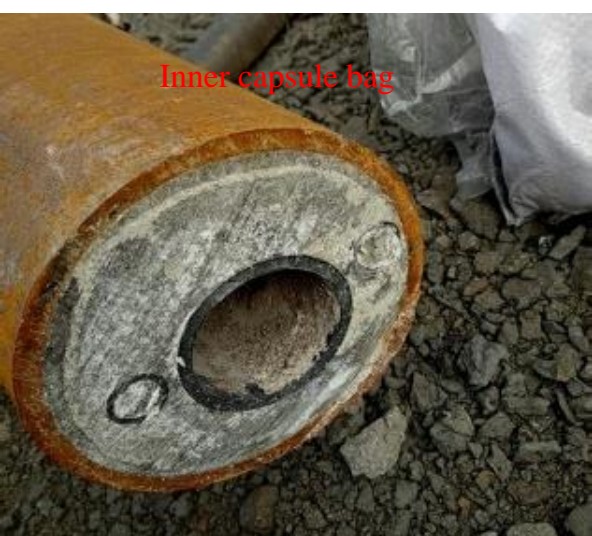

**Figure 15.** Borehole profile of boreholes for "Two plugs, one injection and one row".

Figure 14 shows the part of the traditional "two plugs and one injection" device that is not filled with grout. This part is in the shape of a three-quarter circle. The reason is partly because of cutting jitter and partly because the upward and downward drilling angles are 37°. Figure 15 shows a cross-sectional view of the "two plugs, one injection and one row" device. The bottom and upward borehole grouting part is dense, the grout is completely filled with the plugging section, and there is no air leakage channel, illustrating that the grouting effect of the new process of "two plugs, one injection and one row" is obviously better than the traditional method of "two plugs and one bet".

(2) Analysis of engineering test results

The lower side of 24,130 machine lane is constructed with 10 holes, the sealing depth is 20 m, and the sealing length is 8 m. The downward hole was sealed along the layer and the inclination angle was 25° with all adopting the technology of full screen mesh pipe. Hole 1–5# is grouted and sealed according to the new process of "two plugs, one injection and one row" and hole 1′–5′# was sealed according to the existing two plugs and one injection. After the sealing was completed, 10 pairs of two boreholes were connected to the extraction pipe, and the gas concentration was measured. Figure 16 below shows the average gas concentration of two boreholes with different sealing methods in 90 days.

Through the similar simulation cutting experiment of the uphole plugging device, it can be obtained that the traditional "two plugs and one injection" has a part that is not filled with grout. This part is in the shape of a three-quarter circle. The "two plugs, one injection and one row" borehole grouting is partly dense; the grouting effect of the new "two plugs, one injection and one row" process is significantly better than the traditional "two plugs and one injection" method". Through underground engineering tests, it can be obtained that the average gas concentration of "two plugs, one injection and one row" within 90 days is 75%, and the gas concentration of traditional sealing technology is 55%, which is an average increase of 36.4% compared with traditional technology.

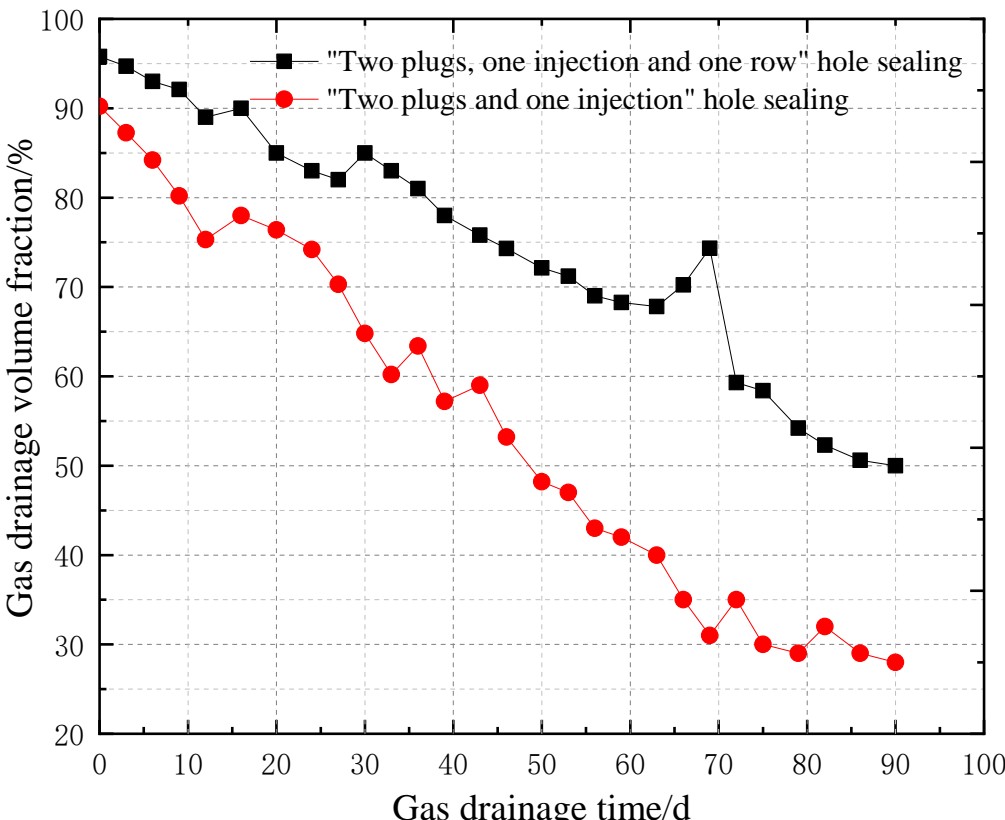

**Figure 16.** Comparison of gas volume fraction of two sealing methods.

### 4. Conclusions

In view of the problem that it is difficult to control the spontaneous combustion of coal around the borehole induced by gas drainage, the judgment criteria for a dangerous area of spontaneous combustion of fractured coal around the borehole were theoretically determined. The combined methods of experiment and simulation were presented in this paper. The comprehensive prevention and control technologies of shotcrete, sealing parameters, and the "two plugs, one injection and one row" device were introduced, resulting in the following conclusions:

(1) The critical oxygen concentration of coal spontaneous combustion was experimentally studied. A method of comprehensively dividing the spontaneous combustion risk area of fractured coal around the borehole induced by gas drainage was proposed based on the oxygen concentration and air leakage speed. The spontaneous combustion risk area was divided by the oxygen concentration of coal around the borehole at8% and the air leakage speed of 0.004 m/s. Scientific division of "dangerous areas of drilling holes" can enhance the pertinence of fire-fighting measures, improve the effect of fire-fighting projects, and effectively prevent spontaneous fire accidents. Production has very important practical significance.

(2) In view of the problem that it is difficult to control the spontaneous combustion of coal around the borehole induced by gas drainage, the optimal roadway shotcrete length was determined to be 2–4 m using the numerical simulation method. The sealing hole depth of 20 m and length of 13 m were determined by combining theory and experiment, which provides a scientific basis for preventing and controlling the spontaneous combustion of coal around the borehole.

(3) On the basis of the "two plugs and one injection" process, the "two plugs, one injection and one row of hole sealers" method was developed. The simulated drilling section was cut. The new process has a compact structure in the pocket part. The traditional process has a three-quarter circular blank section. The gas concentration of the traditional

sealing process is 55% and the average gas concentration of the "two plugs, one injection and one row" method that accumulated within 90 days is 75%, which shows an average increase of 36.4% compared to the traditional process. The new "two plugs, one injection and one row" device is an effective method for improving gas drainage efficiency and preventing spontaneous combustion of gas drainage boreholes.

**Author Contributions:** Conceptualization, J.L. and Y.Z.; Methodology, J.D.; software, J.D. and Y.Z.; validation, J.L., J.L. and J.D.; formal analysis, Y.Z.; investigation, Y.Z.; resources, Y.Z.; data curation, J.L.; writing—original draft preparation, J.L.; writing—review and editing, J.L. and Y.Z.; visualization, J.D. and Y.Z.; supervision, J.D. and Y.Z.; project administration, J.D., J.L. and Y.Z.; funding acquisition, J.L. and J.D. All authors have read and agreed to the published version of the manuscript.

**Funding:** This work was supported by the Fundamental Research Funds for the Central Universities of China (No.3142020020), China Coal Technology & Engineering Group Co., Ltd. (2019-2-ZD003), the National Natural Science Foundation Youth Project of China (grant number 51804161), the National Natural Science Foundation of China (grant number 52074156), the Langfang City Science and Technology Support Plan Project (grant number 2020011017), and the China Postdoctoral Science Foundation (2020M680490).

**Data Availability Statement:** The data used to support the findings of this study are enclosed with the article.

**Acknowledgments:** The authors would like to thank all the reviewers who participated in the review and Rodrigo Cabanero for linguistic assistance during the preparation of this manuscript.

**Conflicts of Interest:** The authors declare no conflict of interest.

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
