# Peer review of "Prevention Technology of Coal Spontaneous Combustion Induced by Gas Drainage in Deep Coal Seam Mining"

_fire, doi:10.3390/fire5030065_

Round 1

Reviewer 1 Report

This work is meanful for developing prevention technology of coal sponta-1 neous combustion induced by gas drainage in deep coal sea, but this  work should be improved in ananlysis and writing. The detail comments are attached.

Reviewer 2 Report

I reviewed the article "Investigation of technology for preventing spontaneous combustion of coal during degassing in a deep coal seam on the example of longwall 24130 of the Pingdingshan 10 mine." The work of the authors is relevant and interesting. 

The authors of the article raise vital research questions. 

But despite the interesting work and the current topic of research, I have several recommendations for its improvement.  

  1. On line 198, the beginning of a sentence begins with a small letter. Needs to be corrected. 
  1. The paper talks about modeling, but I did not see the model itself. Please clarify what model was used? 
  1. I think that for such an interesting article, the review should be improved. I recommend to look at other works of the authors and strengthen your own work. 

I think that the authors did a good job on their research. After a slight improvement of the article, it can be published in the journal. 

Reviewer 3 Report

This study is about the “Research on prevention technology of coal spontaneous combustion induced by gas drainage in 2 deep coal seam: take 24130 working face of Pingdingshan 10 mine as an example”. The following comments emanates from the report:

Title:

The title of the report is too long, and it needs to be shortened.

Abstract:

The abstract is not written well, contains very long sentences and fails to clearly state what was achieved. Clear statements of the novelty of the work should also appear briefly in the Abstract and Conclusions sections.

Introduction:

The main objective of the paper must be written in a more clear and concise manner at the end of the Introduction section. The research gap and how it was addressed should be stated more clearly.

Literature review for spontaneous combustion/self-heating of coal has been gathered in the references. However, the most recent articles with useful information related to this manuscript have been neglected or omitted.

Conclusions

After reading the manuscript, I am still not sure how the conclusions of this manuscript can be used by the coal mining industry. The benefits offered to the industry should be explained clearly. The conclusion section is also missing some perspective related to the future research work. The reference list needs to be reviewed as I see some errors in the reference list. Furthermore, there are a few mistakes in grammar; it is advisable that the overall report should be reviewed by a professional for readability, grammar, and wording before resubmission.

Round 2

Reviewer 1 Report

no comments

Author Response

Dear Reviewer:

Thank you for all your affirmation and valuable comments on this paper. All authors of this article have read your comments carefully and are inspired. We have revised the article and hope to improve it further to your satisfaction.

Reviewer 3 Report

I would like to thank the authors for trying to address my concerns. However, there are still some outstanding issues in the manuscript:

  1. The authors still need to address the research gap so that the manuscript can make sense.
  2. Literature review for spontaneous combustion/self-heating of coal has been gathered in the references. However, the most recent articles with useful information related to this manuscript have been neglected or omitted. I would suggest that the authors should study the following articles and identify more relevant references:
  • Eroglu, H. N., 1992. Factors affecting spontaneous combustion liability index. Ph.D. Thesis, University of the Witwatersrand Johannesburg, South Africa. Yuan, L., and A. Smith. 2008. Numerical study on effects of coal properties on spontaneous heating in longwall gob areas. Fuel 87:3409–19. doi:10.1016/j.fuel.2008.05.015. Nimaje, D. S., and D. P. Tripathy. 2016. Characterization of some Indian coals to assess their liability to spontaneous combustion. Fuel 163:139–47. doi:10.1016/j.fuel.2015.09.041. Onifade, M., Genc, B., 2020. A review of research on spontaneous combustion of coal. Int. J. Min. Sci. Technol. 30 (3), 303–311.
  1. Although the authors improved the Conclusion, it is still missing some perspective related to the future research work. The benefits offered to the coal mining industry should be explained clearly.
  2. The English used in the manuscript requires some proofreading.  
